# Adversarially Robust Imitation Learning

**Jianren Wang***
jianrenw@andrew.cmu.edu

**Ziwen Zhuang***
zhuangzw@shanghaitech.edu.cn

**Yuyang Wang**
yuyangw@andrew.cmu.edu

**Hang Zhao**
hangzhao0124@gmail.com

**Abstract:** Modern imitation learning (IL) utilizes deep neural networks (DNNs) as function approximators to mimic the policy of the expert demonstrations. However, DNNs can be easily fooled by subtle noise added to the input, which is even non-detectable by humans. This makes the learned agent vulnerable to attacks, especially in IL where agents can struggle to recover from the errors. In such light, we propose a sound Adversarially Robust Imitation Learning (ARIL) method. In our setting, an agent and an adversary are trained alternatively. The former with adversarially attacked input at each timestep mimics the behavior of an online expert and the latter learns to add perturbations on the states by forcing the learned agent to fail on choosing the right decisions. We theoretically prove that ARIL can achieve adversarial robustness and evaluate ARIL on multiple benchmarks from DM Control Suite. The result reveals that our method (ARIL) achieves better robustness compare with other imitation learning methods under both sensory attack and physical attack.

**Keywords:** Imitation Learning, Adversarial Learning

## 1 Introduction

Imitation learning is a powerful and practical alternative to reinforcement learning for learning sequential decision-making policies. Together with the computational advancement brought by deep learning, imitation learning has shown great success in autonomous driving [1, 2], robot control [3], game AI [4], and motion capture [5]. Despite achieving human-level performance on many tasks, the existence of adversarial examples [6] in DNNs and many successful attacks to sequential decision-making policies [7, 8, 9, 10] motivate us to study robust imitation algorithms.

The observations by the agent unavoidably contain uncertainty that naturally originates from sensor errors or equipment inaccuracy. A policy not robust to such uncertainty can lead to catastrophic failures (e.g., a small piece of white scotch tape on a traffic sign could cause an improper left turn, which will further lead to miles of unnecessary travel and even crush of the vehicle in the worst case). Therefore, studying the fundamental principles of robust imitation learning is crucial.

One effective approach to induce robustness is domain randomization [11], a method where a designer with expertise identifies the components of the model that they are uncertain about. A set of training environments is constructed where the uncertain components are randomized, ensuring that the agent is robust on average to this set. However, this requires careful parametrization of the uncertainty set as well as hand-designing of the environments. A more easily automated approach is to formulate the problem as a zero-sum game and learn an adversary that perturbs the transition dynamics [12, 13, 14]. If a Nash equilibrium of this problem is found, then that equilibrium provides a worst-case performance bound under the specified set of perturbations. Specifically, we can search over adversary policies $f$ and agent policies $\pi$, so that $\min_{\pi \in \Pi} \max_{f \in \mathcal{F}} J(\pi, f)$, where $J(\pi, f)$ is the expected cumulative reward for the attacker policy. Unfortunately, in most imitation learning settings, quantitative reward from the environment is unavailable.

In this work, we propose a theoretically grounded framework of adversarially robust imitation learning (ARIL), where we build a surrogate loss to measure the distance between the actions generated from

5th Conference on Robot Learning (CoRL 2021), London, UK.

$\pi$ and the actions from the expert. We consider two ways of attack: sensory attack and physical attack. In the sensory attack setting, the adversary only modifies the inputs to $\pi$ (e.g., the underlying true state of the environment is untouched as $s_t$), while in the physical attack setting, the adversary directly modifies the actual physical state, i.e., it changes the underlying state to $s'_t$. We consider attacking under interactive imitation learning (IL) setting where the expert is available for queries at any time step [15]. We evaluate our method in DM Control Suite [16], (e.g. HalfCheetah, Swimmer, Ant), under both sensory attack and physical attack settings. Our method significantly improves the robustness under adversarial attacks over the baseline methods.

To summarize, this paper makes four contributions: (1) We study the theory and practice of Adversarially Robust Imitation Learning (ARIL). (2) We systematically define sensory attack and physical attack. (3) We prove that ARIL can achieve the adversarial robustness under realizability assumption, robust expert assumption, and optimal adversary assumption. (4) We show the efficiency of our method in both attack stage and defense stage on multiple DM Control Suite benchmarks in comparison with other adversarial attack methods and other robust imitation learning methods.

## 2  Related Work

**Adversarial Attack**  Adversarial attacks can be divided into white-box attacks and black-box attacks. In white-box attacks, the attack has access to the structure, algorithm, and system parameters of the target model. In black-box attacks, the attack cannot obtain information from the target model. Instead, it can only interact with the target model by querying with the input and observing the output. In our setting, we consider white-box attacks. Szegedy *et al.*[17] first pointed out the intriguing property of neural networks and proposed a method called Large-BFGS to find adversarial samples. Goodfellow *et al.*[18] proposed a Fast Gradient Sign method (FGSM) to generate small perturbations to fool the neural networks by calculating the gradient of the loss function concerning the inputs. Kurakin *et al.*[19] proposed Targeted FGSM to generate adversarial samples towards a target class and Iterative FGSM to apply FGSM multiple steps with small step sizes. Tramer *et al.*[20] proposed a randomized single-step attack to escape the non-smooth vicinity of the data point. Kurakin *et al.*[19] apply adversarial training on large-scale datasets. Huang *et al.*[7] showed that adversarial examples generated from FGSM could degrade test-time performance of a well-trained NN policy in RL. Sun *et al.*[21] studied the safe sequential decision-making problem under the setting of adversarial contextual bandits which is a one-step RL problem. Lin *et al.*[9] considered strategically-timed attack where the attacker picks specific time steps to attack. Han *et al.*[22] considered attacking an RL algorithm during training time. All these previous works mainly focus on applying attacks developed from the supervised learning literature to RL setting, while in this work, we focus on learning adversarial attacks in imitation learning.

**Robust Imitation Learning**  Imitation learning aims to learn sequential decision-making policies from expert demonstrations. However, the agent can be vulnerable to cascading failures when the trajectory diverges from the demonstrations [23]. Laskey *et al.*[24] proposes to inject noise into the supervisor's demonstration policies, which forces the supervisor to demonstrate how to recover from errors. Wang *et al.*[25] utilizes VAEs [26] and GAIL [27] to learn semantic policy embeddings in imitation learning, which learns robust diverse gaits with limited demonstrations. For robust imitation learning with different-quality demonstrations, Tangkaratt *et al.*[28] makes use of a probabilistic graphical model to estimate the quality of demonstrations. Brown *et al.*[29] proposes a safe and robust Bayesian reward learning through preferences over imitating demonstrations. Shin *et al.*[30] performs the derivative-free random search optimization method with linear policies. In our work, we focus on the robust imitation learning against adversary attacks, which can cause catastrophic failure to DNN-based agents with trivial perturbations.

## 3  Preliminaries

Before we delve into the details of ARIL, we first outline our terminology. In the rest of the paper, we focus on finite-horizon Markov Decision Process (MDP), which is defined as $(\mathcal{S}, \mathcal{A}, H, P, r)$ where $\mathcal{S}$ is the state space, $\mathcal{A}$ is the action space, $r$ is a reward function, $P$ is the transition and $H \in \mathbb{N}^+$ is the finite horizon. A policy $\pi : \mathcal{S} \rightarrow \Delta(\mathcal{A})$ is a mapping that maps from states to distributions over action space, i.e., $\pi(a|s)$ is the probability of picking $a$ given $s$.

## 3.1 Sensory Attack vs Physical Attack

For any pair of policy and attack $(\pi, f)$, we can treat the composition of them as a policy: $\pi \circ f : \mathcal{S} \rightarrow \mathcal{A}$, i.e., $a \sim (\pi \circ f)(s)$ is equivalent to first generate stochastic adversarial example $s' \sim f(\cdot|s)$, and then sample action accordingly $a \sim \pi(\cdot|s')$. Denote $d_\pi$ as the average state distribution induced by policy $\pi$. As $\pi \circ f$ can be understood as a policy, $d_{\pi \circ f}$ will be the state distribution induced by $\pi \circ f$.

We first introduce the *sensory attack* setting where the adversary can only modify the sensory inputs of the RL agent, but not the underlying physical state. Denote a trajectory as $\tau = \{s_1, s'_1, a_1, ..., s_H, s'_H, a_H\}$ generated from the composite policy $\pi \circ f$, the likelihood of such trajectory is $\rho_{\pi \circ f}$:

$$\rho_{\pi \circ f}(\tau) = \prod_{t=1}^{H} P(s_t|s_{t-1}, a_{t-1})f(s'_t|s_t)\pi(a_t|s'_t), \tag{1}$$

where we denote $P(s_1) = P(s_1|s_0, a_0)$ as the initial state distribution. Note that here adversary modifies state $s_t$ to $s'_t$ and then the policy $\pi$ only sees $s'_t$.

For *physical attack* setting, the adversary will have the ability to modify the underlying physical states. Again, we denote a trajectory as $\tau = \{s_1, s'_1, a_1, ..., s_H, s'_H, a_H\}$ generated from the composite policy $\pi \circ f$, the likelihood of such trajectory now is:

$$\rho_{\pi \circ f}(\tau) = \prod_{t=1}^{H} P(s_t|s'_{t-1}, a_{t-1})f(s'_t|s_t)\pi(a_t|s'_t). \tag{2}$$

Note the difference between Eq. 2 and Eq. 1, where transition probability is conditioned on attacked state. In physical attack (Eq. 2), the adversary modifies the state $s$ to $s'$, and then the environment actually generates the next state *based on $s'$*, as the underlying physical state is now $s'$. The sensory attack is similar to attacks considered in today's supervised learning setting, where the physical attack is similar to the robust control setting (e.g., a min-max formulation of $H_\infty$ optimal control [31] under physical disturbances).

Note that under both settings, the attacks are sequential: the adversary can affect the learner's decision via corrupting the state $s$, which will, in turn, affect the future states. In supervised learning, an attack on a given sample will not affect the future samples, since samples are i.i.d in supervised learning.

## 3.2 Expert Robustness

For the expert $\pi^e$, we have: $\quad \forall s \in \mathcal{S}, f \in \mathcal{F}, \pi^e(f(s)) = \pi^e(s). \tag{3}$

In practice consider the setting where the expert $\pi^e$ is a human driver who will not be affected by adversarial examples with small perturbations (e.g., human can easily tell a stop sign even with a sticker on it). We later will relax the notion of expert robustness to include the possibility that the expert will be affected by adversarial attacks.

We consider interactive imitation learning setting proposed in [32] where the expert $\pi^e$ is always available during training time. A learner can query the expert at any state $s$ during training to get feedback $\pi^e(s)$.

# 4 Adversarially Robust Imitation Learning

The goal of adversarial robust imitation learning (ARIL) is to learn a policy that can achieve good performance under all possible attacks:

$$\hat{\pi} = \arg\min_{\pi \in \Pi} \max_{f \in \mathcal{F}} J(\pi, f), \tag{4}$$

where $J(\pi, f)$ is the expected total cost of executing policy $\pi$ with examples corrupted by attack $f$:

$$J(\pi, f) = \mathbb{E}\left[\sum_{h=1}^{H} \ell(a_h, \pi^e(s_h))|a_h \sim \pi(f(s_h)), s_{h+1} \sim P(\cdot|s_h, a_h)\right] \text{ (sensory attack) } \tag{5}$$

$$J(\pi, f) = \mathbb{E}\left[\sum_{h=1}^{H} \ell(a_h, \pi^e(s_h))|a_h \sim \pi(f(s_h)), s_{h+1} \sim P(\cdot|f(s_h), a_h)\right] \text{ (physical attack) } \tag{6}$$

We consider a loss function $\ell(a, a')$ that measures the difference between two actions $a$ and $a'$. For instance, $\ell(a, a')$ could be zero-one loss if $\mathcal{A}$ is discrete, or l2 distance between two actions if $\mathcal{A}$ is continuous.

If we can control the long-term prediction error of the learned policy $\hat{\pi}$ under the worst possible adversary,

$$\max_{f \in \mathcal{F}} \mathbb{E}_{s \sim d_{\hat{\pi} \circ f}} \left[ \mathbb{E}_{a \sim \hat{\pi}(\cdot, f(s))} \left[ \ell(a, \pi^e(s)) \right] \right] \leq \delta, \tag{7}$$

then we can upper bound the performance of $\hat{\pi}$ at test time with the existence of an arbitrary adversary $f \in \mathcal{F}$. To see that, let us consider the setting where $\mathcal{A}$ is discrete and $\ell(a, a')$ is the zero-one loss:

$$J(\pi, f) - J(\pi^e) = \mathbb{E}_{s \sim d_{\pi \circ f}, a \sim \pi(f(s))} \left[ Q^\pi(f(s), a) - Q^e(s, \pi^e(s)) \right] \tag{8}$$

$$\leq 2Q_{\max} \mathbb{E}_{s \sim d_{\pi \circ f}, a \sim \pi(f(s))} [\mathbf{1}[a \neq \pi^e(s)]] \tag{9}$$

$$\leq 2Q_{\max} \max_{f \in \mathcal{F}} \mathbb{E}_{s \sim d_{\pi \circ f}, a \sim \pi(f(s))} [\mathbf{1}[a \neq \pi^e(s)]] \tag{10}$$

$$= 2Q_{\max} \max_{f \in \mathcal{F}} \mathbb{E}_{s \sim d_{\pi \circ f}} \left[ \mathbb{E}_{a \sim \pi(f(s))} [\ell(a, \pi^e(s))] \right] \leq 2Q_{\max} \delta. \tag{11}$$

Here, $Q^e$ denotes the state-action value of the expert policy and $Q_{\max}$ denotes the upper bound of $Q^e$. Namely, if $\delta$ is small, the learned policy $\hat{\pi}$ can achieve similar performance as the expert, even under an adversarial attack $f$ during the execution of $\hat{\pi}$. This justifies the reason for finding $\hat{\pi}$ that minimizes the objective function in Eq. 7.

Let us define the objective as follows:

$$\mathcal{L}(\pi, f) = \mathbb{E}_{s \sim d_{\pi \circ f}} \left[ \mathbb{E}_{a \sim \pi(f(s))} \left[ \ell(a, \pi^e(s)) \right] \right] \tag{12}$$

We perform an iteratively alternating update for $\pi$ and $f$.

## 4.1 Update $\pi$ with fixed $f$

At iteration $t$, given the current $f_t$, and $\pi_t$, we want to update $\pi_t$ to $\pi_{t+1}$. We consider the loss function with respect to $\pi$ at round $t$:

$$\ell_t(\pi) = \mathbb{E}_{s \sim d_{\pi_t \circ f_t}} \mathbb{E}_{a \sim \pi(f_t(s))} \left[ \ell(a, \pi^e(s)) \right]. \tag{13}$$

which is a classification loss where features are corrupted sampled $s \sim \tilde{d}_{\pi_t \circ f_t}$, and labels are computed from expert query $\pi^e(s)$.

Note that for discrete action space, the above loss function is simply a classification loss of the classifier $\pi$, with features generated from the distribution $\tilde{d}_{\pi_t \circ f_t}$, and the ground truth label is generated from $\pi^e$. We perform no-regret update on $\ell_t(\pi)$ to generate $\pi_{t+1}$.

$$\pi_{t+1} = \arg \min_{\pi \in \Pi} \frac{1}{t} \sum_{i=1}^{t} \ell_i(\pi). \tag{14}$$

Note that the above optimization can be understood as a classification task where the features are sampled from the mixture of distributions $\{\tilde{d}_{\pi_i \circ f_i}\}_{i=1}^t$ with equal weight $1/t$, and the corresponding labels are generated based on $\pi^e \circ f_i$ for a data point sampled from $\tilde{d}_{\pi_i \circ f_i}, \forall i \in [t]$. In practice, we need to use samples to replace expectations, which leads us to a Data Aggregation style update (DAgger [32]) for $\pi$.

## 4.2 Update $f$ with fixed $\pi$

At iteration $t$, given the current $f_t$ and $\pi_t$, we want to update $f_t$ to $f_{t+1}$. We consider the objective function with respect to $f$ at round $t$ as:

$$\tilde{\ell}_t(f) = \mathbb{E}_{s \sim d_{\pi_t \circ f}} \left[ \ell(\pi_t(f(s)), \pi^e(s)) \right]. \tag{15}$$

We want to find $f$ that maximize the above objective function, namely, increasing the classification loss of the current policy $\pi_t$ as much as possible. Note that different from the loss function for updating $\pi$, here the expectation is defined with respect to the fixed $\pi_t$ and $f$. If $\pi_t$ is a stochastic

---

**Algorithm 1** Adversarial Robust Imitation Learning (ARIL)

---

1: Input: MDP $\mathcal{M}$, policy class $\Pi$, expert policy $\pi^*$, set of adversaries $\mathcal{F}$, a supervised learning algorithm LEANER
2: Initialize a dataset $\mathcal{D} = \emptyset$
3: Initialize a policy $\pi_0 \in \Pi$
4: **for** $t = 0$ to $N$ **do**
5:      $f_t = \arg\max_{f \in \mathcal{F}} \mathbb{E}_{s \sim d_{\pi_t \circ f}} [\ell(\pi_t(f(s)), \pi^e(s))]$ (RL Step)
6:      Execute $\pi_t \circ f_t$ to generate a set of non-corrupted states $\{s_i\}_{i=1}^n$
7:      Query expert during execution to get labels $\{\pi^e(s_i)\}_{i=1}^n$
8:      Data Aggregation: $\mathcal{D} = \mathcal{D} + \{f_t(s_i), \pi^e(s_i)\}_{i=1}^n$
9:      $\pi_{t+1} = \text{LEARNER}(\mathcal{D})$ (Supervised Learning, e.g., Classification if $\mathcal{A}$ is discrete )
10: **end for**

---

policy, and $f$ is parameterized as $f(s; \theta)$, then one can use policy gradient technique to compute the $\nabla_\theta \tilde{\ell}_t(f)$ as follows:

$$\nabla_\theta \tilde{\ell}_t(f(\cdot|\theta))|_{\theta_t} = \nabla_\theta \mathbb{E}_{\tau \sim \rho_{\pi_t \circ f}} \sum_{h=1}^H \ell(a_h, \pi^e(s_h)) = \mathbb{E}_{\tau \sim \rho_{\pi_t \circ f_{\theta_t}}} \left[ \nabla_\theta (\ln \rho_{\pi_t \circ f}(\tau)) \sum_{h=1}^H \ell(a_h, \pi^e(s_h)) \right] \tag{16}$$

$$= \mathbb{E}_{\tau \sim \rho_{\pi_t \circ f_{\theta_t}}} \left[ \sum_{h=1}^H \nabla_s \ln \pi_t(a_h|s)|_{f(s_h; \theta_t)} \nabla_\theta f(s_h; \theta)|_{\theta_t} \left( \sum_{h'=h}^H \ell(a_h, \pi^e(s_h)) \right) \right] \tag{17}$$

$$= \sum_{h=1}^H \mathbb{E}_{s_h \sim d_{\pi_t \circ f_t}^h, a_h \sim \pi_t(f_t(s_h))} \left[ \nabla_s \ln \pi_t(a_h|s)|_{f(s_h; \theta_t)} \nabla_\theta f(s_h; \theta)|_{\theta_t} Q^{\pi_t \circ f_t}(s_h, a_h) \right]. \tag{18}$$

where the second step simply uses chain rule, and in the last step we simply treat $\pi \circ f$ together as a policy that maps from states to actions—hence the Q function is defined with respect to $\pi_t \circ f_t$. Intuitively, we want to optimize $f_\theta$ such that it increases the likelihood of generating actions via $\pi_t$ to increase the total "reward"—the long-term prediction error (with respect to $\pi^e$) of the policy $\pi_t$. Using the above gradient to perform gradient ascent to optimizing $f_\theta$ is similar to classic policy gradient methods and in practice, one can use existing techniques developed for PG here as well (e.g., natural gradient, actor-critic where the critic is $Q^{\pi_t \circ f_t}$).

### 4.3 Algorithm

We summarize the algorithm ARIL in Alg. 1, where every round, we update the adversary $f$ and the policy $\pi$. Given a policy $\pi_t$, we aim to find the adversary $f_t$ from $\mathcal{F}$ that increases the long-term prediction error of $\pi_t$ (w/ expert's prediction) as much as possible. Once we find such adversary $f_t$, we generate data by executing the composite policy $\pi_t \circ f_t$ on the real system to generate states $\{s_i\}_{i=1}^n \sim d_{\pi_t \circ f_t}$ and labels $\{\pi^e(s_i)\}_{i=1}^n$ (this procedure requires interactively querying expert, just as DAgger does [32]). We then corrupt the data by applying the latest adversary $f_t$ at every data point $s_t$, and appendix the new dataset $\{f_t(s_i), \pi^e(s_i)\}_{i=1}^n$ to the aggregated dataset $\mathcal{D}$. Lastly, we train the new policy using the black box supervised learner on the aggregated dataset $\mathcal{D}$.

The data aggregation technique (DAgger [32]) is used to ensure stability in the process of updating $\pi$ and corresponds to the specific no-regret learner FTL as we explained before. However, any other no-regret learner can be used here as well, such as Online Gradient Descent [33], with the potential benefit of no need to maintain an ever-growing dataset $\mathcal{D}$.

In theory, optimizing $f_t$ is challenging as the objective function could potentially be non-convex here. However, in practice for parameterized adversary $f$, as we showed in the previous section, we can compute the gradient of $f$ using classic policy gradient technique, which in turn allows us to leverage practically effective techniques such as natural gradient, actor-critic, to optimize $f_t$. Note that the optimization procedure for $f_t$ does not require any expert data.

## 4.4 Analysis

We consider the following setting. First, for optimization in $f_t$, we assume we can achieve near-optimality:

$$\tilde{\ell}_t(f_t) \geq \max_{f \in \mathcal{F}} \tilde{\ell}_t(f) - \epsilon_{rl}, \tag{19}$$

with $\epsilon_{rl} \in \mathbb{R}^+$. Namely, we do not assume we can find the global optimal adversary at every round. The $\epsilon$ here quantifies the performance of the potential RL algorithm in terms of optimizing $\tilde{\ell}_t(f)$ with respect to $f$.

For simplicity, we consider the *realizability setting*: there exists a policy $\pi \in \Pi$, such that for any state $s \in \mathcal{S}$, $\mathbb{E}_{a \sim \pi(s)}[a \neq \pi^e(s)] = 0$. Namely there exists at least one policy in policy class $\Pi$ that can predict expert's action at any state $s \in \mathcal{S}$. Relaxing the realizability setting to agnostic setting is straightforward here.

**Theorem 4.1.** *As $T \to \infty$, Alg. 1 outputs a policy $\pi_{i^\star} \in \{\pi_t\}_{t=0}^T$ such that:*

$$\max_{f \in \mathcal{F}} \mathbb{E}_{s \sim d_{\pi_{i^\star} \circ f}} \mathbb{E}_{a \sim \pi_{i^\star} \circ f(s)}[\ell(a, \pi^e(s))] \leq \epsilon_{rl} + \epsilon_e. \tag{20}$$

The above theorem indicates that (1) under the realizability assumption (i.e., policy class $\Pi$ is rich enough to approximate $\pi^e$ well), (2) the assumption that the expert is robust with respect to any attack (i.e., $\epsilon_e$ is defined as the error between $\pi^e(s)$ and $\pi^e(f(s))$, which is negligible), and (3) the assumption that we can near-optimally solve the RL-like optimization problem $\max_{f \in \mathcal{F}} \tilde{\ell}_t(f)$ at every round $t$ (i.e., $\epsilon_{rl}$ is small), then Alg. 1 can achieve adversarial robustness. Please refer to supplementary material for proof.

## 5 Experiments

We evaluate our proposed adversary and robust imitation algorithm with baseline methods using DM Control Suite [16]. We compare our method with FGSM [18]. Considering FGSM is commonly used in supervised learning setting, we apply FGSM to modify state $s_h$ such that it hurts the current *one-step prediction* of $\pi_t$, i.e. $f_{\text{fgsm}}(s_h) = s_h + \epsilon \text{sign}(\nabla_s \mathbb{E}_{a \sim \pi_t(s)}(\ell(a, \pi^e(s_h)))|_{s_h})$.

Our algorithm achieves better robustness under both sensory attack and physical attack. The details of the datasets and other experimental settings are described below.

### 5.1 ARIL with Sensory Attack

In sensory attack, the adversary only modifies the sensory inputs of the agent, but not the underlying physical state, which can be treated as sensory noise in real scenarios.

**Environment Setting**  We investigate the performance of attacks on three continuous control tasks from OpenAI gym robotic benchmarks [34]: *Ant-v2*, *HalfCheetah-v2* and *Swimmer-v2*. For the convenience of training and testing, we set the horizon $H$ to 1000 in all the benchmark environments.

We only attack the zero-order state in the robotics environments, i.e. the positions and angles other than velocities (first-order state). The attack is bounded by $\pm \epsilon$ ($\epsilon$ is a small positive number), which varies according to the robustness of each environment. The $\epsilon$ is chosen to have no more than 40% performance drop with FGSM attack and its values for all experimental setups can be found in supplementary materials. For comparison, all attack methods are bounded to the same $\epsilon$ when applying to the same robotics environment.

**Training Details**  Our implementation of ARIL is built on top of rllab [35] and uses Trust Region Policy Optimization (TRPO) [36]. Throughout the experiments, both policy $\pi$ and adversary $f_\theta$ are parameterized with two hidden layers. The output layer of adversary uses *tanh* as activation function and then multiples $\epsilon$ to restrict the output in $[-\epsilon, \epsilon]$. The standard deviation of $f_\theta$ is initialized to $0.1\epsilon$ and the attack to state $s$ is sampled from $\mathcal{N}(f_\theta(s), 0.1\epsilon)$. We also clip the sampled adversarial noise to $[-\epsilon, \epsilon]$. Hyperparameters are selected by grid search and the values for all experimental setups can be found in supplementary materials.

**Results**  We first compare our adversary with the baseline Fast Gradient Sign Method (FGSM) [18] and Transferable adversarial perturbations (FGSM^k) [37] In both our method and FGSM, the perturbation added to each state is bounded to a small value such that it's imperceptible as shown in the supplementary material.

From Figure 1 (top), we first observe that FGSM attack in imitation learning works well. By forcing the policy to generate actions that are far away from the expert's, FGSM attack is effective and successfully decreases the performance of $\pi$. The performance of FGSM and our attacks in all settings can be found in Table 1 (Adversary). In all tasks, our attack outperforms the FGSM attack. We experimentally demonstrate that through maximizing the total difference to expert policy, the adversarial attack could significantly decrease the total reward. We also show that naively applying adversarial attack techniques developed for supervised learning to a sequential decision-making agent is not effective since the attack is myopic. This result proves the efficiency of our attacks even without ever accessing to reward signals.

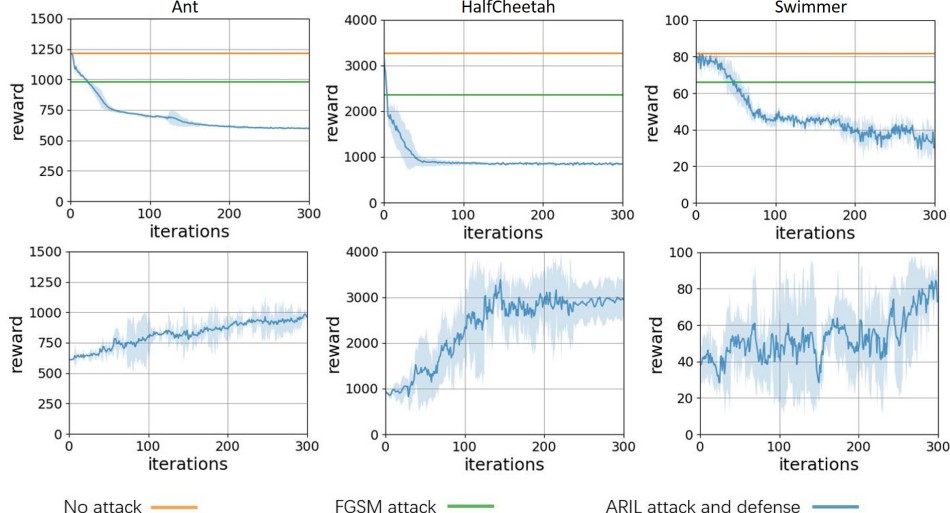

Figure 1: The first row and second row illustrate the results of sensory attacks and defense on sensory attacks, respective, in Ant, HalfCheetah and Swimmer environment.

We then compare ARIL with another robust imitation learning method, DART [24]. DART learns to inject an optimized level of noise into the expert demonstrations in imitation learning, which forces the expert to demonstrate how to recover from the errors. A noise distribution, $p(\xi|\pi_{\theta*}, \psi)$, is injected to the demonstration distribution $\pi_{\theta*}(u|x, \psi)$, where $\psi$ denotes the sufficient statistics that define the noise distribution. In our experiments, we set $\pi_{\theta*}(u|x, \psi)$ as a Gaussian distribution, $\mathcal{N}(\pi_{\theta*}(x), \Sigma)$. The optimal parameter $\hat{\psi}$ is estimated by the anticipated final robot's error and subsequently scaling the current simulated error to this level, denoted as $\mathbb{E}_{p(\xi|\pi_{\theta*}, \hat{\psi})} \sum_t \ell(u_t, \pi_{\pi*}(x_t))$. For quantitative comparison, we train an adversary to attack DART using our proposed attacker (ARIL-A). As shown in Table 1 (bottom), our method significantly improves the robustness over DAgger [32] (3rd row and 5th row). ARIL also achieves better robustness compared with DART (4th row and 5th row) in all tasks under the worst possible adversary.

Table 1: Performance of various sensory attack and robust imitation learning methods

|  |  | Ant-v2 | HalfCheetah-v2 | Swimmer-v2 | Overall (%) |
|---|---|---|---|---|---|
| Adversary | DAgger [15] | 1,245 | 3,276 | 81 | 0% |
|  | FGSM [18] | 945 (-24.1%) | 2,336 (-28.7%) | 66 (-18.5%) | -23.7% |
|  | FGSM^k [37] | 884 (-28.9%) | 1,973 (-39.8%) | 57 (-29.6%) | -32.7% |
|  | ARIL-A | 607 (-51.2%) | 830 (-74.7%) | 38 (-53.1%) | -59.7% |
| Robust IL | GAIL [27] | 638 (-48.8%) | 1,037 (-68.3%) | 42(-48.1%) | -55.1% |
|  | DR [38] | 679(-45.5%) | 1,372(-58.1%) | 56(-30.9%) | -44.8% |
|  | DART [24] | 779 (-37.4%) | 1,832 (-44.1%) | 49 (-39.5%) | -40.3% |
|  | ARIL | 965 (-22.5%) | 2,974 (-9.2%) | 72 (-11.1%) | -14.3% |

## 5.2 ARIL with Physical Attack

In physical attack, the attacker can also affect the underlying transition dynamics. Considering the attacker changes the actual state of in the environment, the transition model can be described as

$s_{t+1} = f(P(\cdot|s_t, a_t))$. Namely, the actual zero-order states are changed under the attack. We can treat physical attack as an external perturbation (e.g. a kick on the robot or an unpredicted bump on the road).

**Environment Setting**    Similar to sensory attack, the attack is bounded by $\pm\epsilon$. It's worth mentioning that $\epsilon$ is different for sensory attack and physical attack even in the same environment. Since physical attack with the ability to change underlying dynamics is more effective than sensory attack. The $\epsilon$ are chosen to have no more than 40% performance drop with FGSM attack and its values for all experimental setups can be found in supplementary materials.

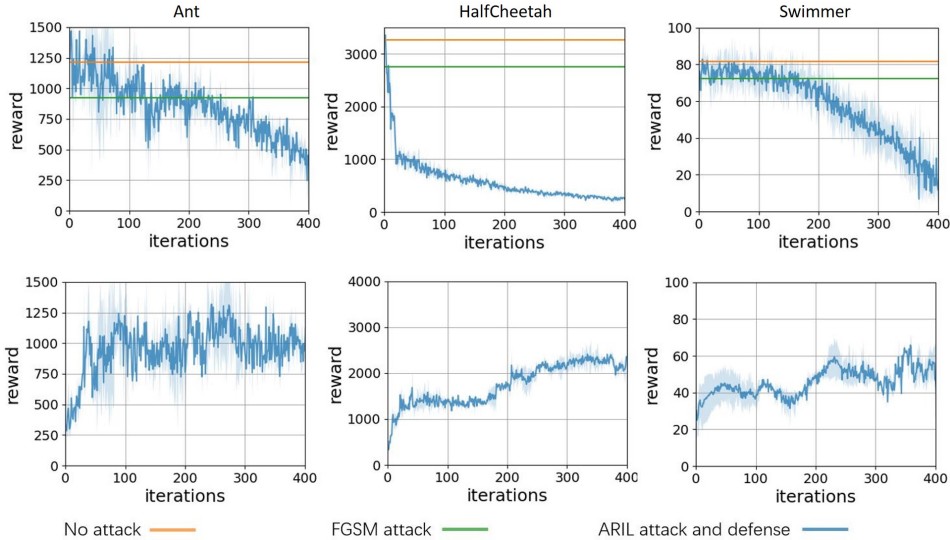

Figure 2: The first row and second row illustrate the results of physical attacks and defense on physical attacks, respective, in Ant, HalfCheetah and Swimmer environment.

**Results**    We first compare our adversary with the FGSM baseline. As one might expect, the physical attack is more effective than the sensor attack. Under the physical attack setting, $\pi$ is more likely to fail. Our experiments support this viewpoint in the sense that $\epsilon$ for sensory attack needs to be larger than the one for the physical attack to achieve similar attack performance. Again, in all tasks, our attack outperforms the FGSM attack. We also compare ARIL with the DART (Table 2). This result proves the efficiency of our robust imitation algorithm even the underlying transition dynamics are changed.

Table 2: Performance of various physical attack and robust imitation learning methods

|  |  | Ant-v2 | HalfCheetah-v2 | Swimmer-v2 | Overall (%) |
|---|---|---|---|---|---|
| Adversary | DAgger [15] | 1,245 | 3,276 | 81 | 0% |
|  | FGSM [18] | 885 (-28.9%) | 2,100 (-35.9%) | 61 (-24.7%) | -29.8% |
|  | ARIL-A | 158 (-87.3%) | 249 (-92.5%) | 16 (-80.2%) | -86.7% |
| Robust IL | DART [24] | 581 (-53.3%) | 1,078 (-67.1%) | 27 (-66.7%) | -62.4% |
|  | ARIL | 903 (-27.5%) | 2,183 (-33.4%) | 56 (-30.1%) | -30.3% |

# 6   Conclusion

In conclusion, we propose an adversarially robust imitation learning algorithm. We theoretically prove that under realizability assumption, robust expert assumption, and optimal adversary assumption, ARIL achieves adversarial robustness. We evaluate our method on a set of continuous control tasks from OpenAI Gym and show that our method achieve better robustness under both sensory attack and physical attack compare with baselines. We hope that our works can inspire more studies in robust imitation learning methods when the quantitative reward from the environment is unavailable.

**Acknowledgements.** The authors would like to thank Wen Sun, Yan Xu, Yunze Man for fruitful discussion and detailed feedback. This work was supported by the PanGU Young Investigator Award.

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

# A  Proof of Theorem 4.1

*Proof.* Since Alg. 1 runs no-regret online learner to update $\pi$ on the sequence of loss functions $\{\ell_t(\pi)\}$, we must have:

$$\sum_{t=0}^{T} \ell_t(\pi_t) - \min_{\pi \in \Pi} \sum_{t=0}^{T} \ell_t(\pi) \leq o(T). \tag{21}$$

Add and subtract $\sum_{t=0}^{T} \ell_t(\pi^e)$ on the left hand side of the above inequality, we have:

$$\sum_{t=0}^{T} \ell_t(\pi_t) - \sum_{t=0}^{T} \ell_t(\pi^e) \leq o(T) + \left( \min_{\pi \in \Pi} \sum_{t=0}^{T} \ell_t(\pi) - \sum_{t=0}^{T} \ell_t(\pi^e) \right) \tag{22}$$

Since we operate under the realizability setting, the term inside the parenthesis on the RHS of the above inequality is guaranteed to be less than or equal to zero. Hence, the above inequality simplifies to:

$$\sum_{t=0}^{T} \ell_t(\pi_t) \leq \sum_{t=0}^{T} \ell_t(\pi^e) + o(T). \tag{23}$$

For $\ell_t(\pi^e)$, using the definition of $\ell_t$, we see:

$$\ell_t(\pi^e) = \mathbb{E}_{s \sim d_{\pi_t \circ f_t}} \left[ \mathbb{E}_{a \sim \pi^e(f_t(s))}[\ell(a, \pi^e(s))] \right] = \mathbb{E}_{s \sim d_{\pi_t \circ f_t}} \left[ \ell(\pi^e(f_t(s)), \pi^e(s)) \right] = 0, \tag{24}$$

where we use the expert robustness definition above (i.e., $\pi^e(f(s)) = \pi^e(s)$ for all $s$ and $f$). Hence, we have:

$$\sum_{t=0}^{T} \ell_t(\pi_t) \leq T\epsilon_e + o(T). \tag{25}$$

Now we need to lower bound $\ell_t(\pi)$. Using the definition of $\ell_t$ again, we have:

$$\max_f \mathcal{L}(\pi_{i^\star}, f) \leq \mathcal{L}(\pi_{i^\star}, f_{i^\star}) + \epsilon_{rl} \leq \frac{1}{T} \sum_t \mathcal{L}(\pi_t, f_t) + \epsilon_{rl} = \frac{1}{T} \sum_t \ell_t(\pi_t) + \epsilon_{rl}, \tag{26}$$

where the first inequality comes from the assumption that the RL solver returns a $\epsilon_{rl}$ near-optimal solution.

Combine all the results above together, we get:

$$\max_f \mathcal{L}(\pi_{i^\star}, f) \leq \epsilon_{rl} + \epsilon_e + o(T)/T. \tag{27}$$

Hence as $T$ approaches to $\infty$, we have that the long-term prediction loss of the learned policy $\pi_{i^\star}$ under the worst possible adversarial attack from $\mathcal{F}$ is upper bounded by $\epsilon_{rl}$—the error introduced from optimizing $f_t$.

Under agnostic setting, it is not guaranteed that there exists $a = \pi^e(s)$, where $a \sim \pi(s)$ and $\pi \in \Pi$ for $\forall s \in \mathcal{S}$. However, it can be assumed that the error between $a$ and $\pi^e(s)$ is bounded by a small number $\epsilon_a$, namely $\mathbb{E}_{a \sim \pi(s)}[a \neq \pi^e(s)] \leq \epsilon_a$. Hence Equation 4 is modified as:

$$\ell_t(\pi^e) = \mathbb{E}_{s \sim d_{\pi_t \circ f_t}} \left[ \mathbb{E}_{a \sim \pi^e(f_t(s))}[\ell(a, \pi^e(s))] \right] \leq \epsilon_a. \tag{28}$$

Plugging this to Equation 3, we have:

$$\sum_{t=0}^{T} \ell_t(\pi_t) \leq T(\epsilon_e + \epsilon_a) + o(T). \tag{29}$$

Combining the results with Equation 6, we get

$$\max_f \mathcal{L}(\pi_{i^\star}, f) \leq \epsilon_{rl} + \epsilon_e + \epsilon_a + o(T)/T. \tag{30}$$

Similarly to the realizability setting, as $T$ approaches to $\infty$, the long-term prediction loss of the learned policy $\pi_{i^\star}$ under the worst possible adversarial attack from $\mathcal{F}$ is still upper bounded.

$\square$

### A.1 Training Details

Both attacker and student are trained using Adam optimizer at learning rate of $0.001$.

At the start of ARIL algorithm, we initialized the buffer for DAgger algorithm with size $5M$ timesteps. In each attack stage, it takes $1e6$ timesteps for the attacker to collect the trajectories each time. Then at each defense stage, we collected $40$ trajectories of student under attack and labeled them with expert actions into the DAgger buffer.

Table 3: Hyperparameters for training attacker

| | |
|---|---|
| entropy loss coefficient | 0.01 |
| value loss coefficient | 0.5 |
| clip range | 0.2 |

