# OpenReview forum: "Adversarially Robust Imitation Learning"
_robot-learning.org/CoRL/2021/Conference — CoRL2021 Poster_

### Official Review · Reviewer_1bf5 · 2021-07-19

**Originality:** Good
**Technical Quality:** Good
**Clarity Of Presentation:** Fair
**Impact:** 4

**Recommendation:**

Strong Accept: I recommend accepting the paper and will argue for my recommendation even if other reviewers hold a different opinion.

**Summary:**

The paper proposes an adversarially robust imitation learning (ARIL) method which learns an MDP policy that is robust against state perturbation. This is achieved by iteratively updating a policy to minimize the expected cost and an adversary to maximizes the expected cost under the data aggregation (Dagger) framework. The paper proofs that under certain assumptions, the worst-case performance of the policy against the best adversarial is bounded. ARIL is empirically evaluated against an adversarial attack method (FGSM) and an adversarial defense method (DART) on continuous-control tasks.

**Issues:**

Besides the issues in experiments commented above, the following are mathematical inconsistency and ambiguity that should be addressed in the revision.
- Line 36 defines the optimization problem $\min_{\pi} \max_{f} J(\pi, f)$ with the expected reward $J$. This has to be a typo since the policy should be maximizing the expected reward instead of minimizing it.
- The paper formulates the problem with an MDP with the reward $r$, but later formulates the policy learning problem using the cost $c$. While it can be understood that $c(s,a) = -r(s,a)$, this inconsistency should be addressed for unfamiliar readers.
- Equation 7 contains the difference $Q^e(s,a) - Q^e(s,\pi^e(s))$, but the first Q-value should be $Q^\pi(s,a)$ because it comes from the expected return of the learner $J(\pi, f)$. In addition, the Q-value and the constant $Q_{\mathrm{max}}$ is not defined anywhere. While these quantities can be often understood for the context, they should be formally defined in the paper.
- Equation 13 should have $\ell_i(\pi)$ instead of $\ell_t(\pi)$ otherwise the summation operator should be removed.
- Equations 16-17 contains the joint distribution of state-action: $\pi_t(a_h,s)$. I think it should be the conditional distribution $\pi_t(a_h|s)$ instead. Otherwise, the gradient is not computable since the joint is unknown and you require the unknown state marginal to compute it using the policy.
- The error $\epsilon_e$ in the theorem is not defined anywhere, even though it is claimed to be small when the expert is robust. It is also not defined in the appendix. In fact, $\epsilon_e$ suddenly appears in equation 5 of the appendix without explanation.


**Reviewer Expertise:**

Very good: Comprehensive knowledge of the area

**Strengths And Weaknesses:**

### Strength
- The paper proposes a promising method for robust imitation learning under state perturbation. The results of this paper should be useful for robust imitation-learning research in the future.
- The paper proposes both attack and defense mechanisms and evaluates both of them in the experiments. However, this point is expanded only in the experiments. I suggest to include this point into the discussions in early sections.
- The method is supported by a theoretical result showing that the performance of learned policy can be bounded. Still, the result assumes the zero-one loss which is not used in practice. I suggest expanding the results to consider surrogate losses instead of the zero-one loss.

### Weakness
- There are too few baseline methods in the experiments. The paper should include more methods, especially defensive methods, in the experiments.
- The paper does not evaluate the method in a physical robot with real-world state perturbation. While this is not a major issue, the paper would be much more convincing with a result on a physical robot.
- There are mathematical inconsistency and ambiguity, possibly due to typos. The details of these are commented in the Issues section below.

**Summary Of Recommendation:**

The paper proposes an interesting and promising approach to make imitation learning more applicable to real-world scenarios. While the technical part of the paper is not very new (i.e., the method still follows standard min-max optimization in robust learning), it is a new idea to apply the technique to imitation learning. The derived performance-bound is also a strong addition to the contribution. The major issue of the paper to me is the experiment part, which demonstrates the method only in simulated robots but not in physical robots. The mathematical clarity is also a minor issue, but this can be straightforwardly addressed. Overall, I rate to weak accept the paper.

---

### Official Review · Reviewer_uSZo · 2021-07-22

**Originality:** Good
**Technical Quality:** Good
**Clarity Of Presentation:** Good
**Impact:** 2

**Recommendation:**

Weak Accept: I recommend accepting the paper, but will not argue for my recommendation if the majority of other reviewers have a different opinion.

**Summary:**

This paper proposed an adversarial robust imitation learning algorithm to improve imitation robustness. Sensory and physical attack are defined and considered in this paper. The authors theoretically proved that the proposed ARIL algorithm can achieve adversarial robustness. Four experimental benchmark results showed the efficiency of this method compared with other methods.

**Issues:**

More experimental results should be added, such as hose hard tasks like humanoid-v2 or self-driving tasks.

**Reviewer Expertise:**

Excellent: Expert knowledge on the topic of the paper

**Strengths And Weaknesses:**

Strengths.
Theory and implementation of this ARIL method are reasonable.
Weakness.
1.	There are only 4 benchmark results from DM control suite. I strongly suggest add more experimental results especially those hard tasks like humanoid-v2, which are more sensitive to uncertainty.
2.	Other imitation learning algorithms should be considered as comparison such as GAIL like methods.


**Summary Of Recommendation:**

As there is no practical tasks such as self-driving to verify the proposed algorithm, it is hard to assess the theory results in this paper, which are based on strong assumptions.

---

### Official Review · Reviewer_bbzz · 2021-07-23

**Originality:** Good
**Technical Quality:** Very Good
**Clarity Of Presentation:** Very Good
**Impact:** 3

**Recommendation:**

Weak Accept: I recommend accepting the paper, but will not argue for my recommendation if the majority of other reviewers have a different opinion.

**Summary:**

The paper presents a method for imitation learning that is robust to both sensor noise and physical disturbances, via training adversarial policies with RL. The method is demonstrated on various standard RL benchmarks.

**Issues:**

I'm not an expert in this area, but it seems quite similar in approach to several other papers.

**Reviewer Expertise:**

Fair: Some knowledge of the area

**Strengths And Weaknesses:**

Strengths
- The paper is well-written and easy to follow
- The method seems reasonable and potentially useful.
- There are some theoretical results confirming robustness under reasonable assumptions.
- The experimental results show benefits over simple methods such as FGSM.

Weaknesses
- This is a minor point, but the authors state "The sensory attack is similar to attacks considered in today’s supervised learning setting, where the physical attack is similar to the robust control setting (e.g., a min-max formulation of H∞ optimal control [31])"
In fact, in H∞ control it is standard to consider worst-case (adversarial) disturbances to both sensors and the true physical state. Actually, the problem posed in the paper is essentially a nonlinear version of standard problems in H∞ control.
- The idea of imitation learning is to approximate the optimal policy, but Sec 4 the authors propose comparing alternative policies via a simple loss such as zero-one or l2. But policies should be compared based on their ultimate effect on the system, e.g. via a Q function approximation. Have the authors considered this?
- The authors compare mainly agains "vanilla" FGSM. But this is a very simplistic algorithm and there are many variations that could perform better, e.g. iterative gradient versions.
- There is some recent work in adversarial RL, e.g. by Russo and Proutiere, which seems similar in spirit. Have the authors compared to this? (Not exactly for imitation learning, but there are similarities)




**Summary Of Recommendation:**

Overall, I thought the paper was quite interesting and presented a potentially useful method.

---

### Official Review · Reviewer_Bpno · 2021-07-23

**Originality:** Good
**Technical Quality:** Good
**Clarity Of Presentation:** Fair
**Impact:** 4

**Recommendation:**

Strong Accept: I recommend accepting the paper and will argue for my recommendation even if other reviewers hold a different opinion.

**Summary:**

This paper considers training an imitation learning agent to be robust to adversarial changes in the observations or actual state. Using a DAgger style algorithm the authors train an agent to follow expert actions which are assumed to be minimally affected by the adversary's changes. The algorithm is shown to work well in terms of robustness and some theory motivating the approach is presented.

**Issues:**

In section 3.2 the authors assume the expert is not affected by adversarial examples, but later in Theorem 4.1 the authors include new notation \eps_e that is undefined and seems to denote that the expert is affected. Please clarify and define assumptions better.

Do you consider this an algorithm that should be used with a human supervisor or an algorithmic supervisor? If human, see below for references on how DAgger is burdensome for a human and how to alleviate this. If algorithmic, how does this method compare to domain randomization applied to imitation learning? There the idea is similar in that you apply different noise and rendering styles to the input images but force the learner to always take the same action regardless of the perturbation. See for example, Seita et al. "Deep Imitation Learning of Sequential Fabric Smoothing From an Algorithmic Supervisor" IROS 2020. It would be good to compare your results with some form of domain randomization approach to show that optimizing for adversarial noise is advantageous.

Clarity / typos
Line 26 -> "damage to the vehicle in the worst case"
Line 106 -> "Again, denote ..."
Line 130: what is this notation? Is it supposed to be an indicator? maybe use mathbf{1}?
Line 135: Q is never defined. I assume it is the state-action value, but this should be clarified in the text. What is Qmax? Can this be arbitrarily large? Is this bound vacuous? Is there an equivalent bound for a non-zero one loss?
Line 149: Is there a reference for Follow the Leader algorithm? I'm not familiar with it.
Line 151: 1/t isn't equal weight since t is increasing, right?
Line 6 in Alg 1 doesn't seem necessary given the assumption that the expert isn't affected by the corruption. This assumption or the relaxatino of it should be clarified
Line 162 and elsewhere: notation is confusing, sometimes uses a and sometimes uses a_h. In eq (15) a_h is undefined. Also s in (16) is not defined should it have a subscript? what is \rho here. What is ln (\rho)? what is Q is it wrt the true cost or the prediction error loss?
Line 196: If the agnostic setting is straightforward then why not do it? At least in the appendix? What is the benefit of the current approach compared to the agnositic setting. It would be good to show the most useful/interesting case in the paper
Line 239: Reword since it seems that FGSM *is* effective, just not as much as the proposed approach.

Equation (2) is identical to (1) except for a single tick mark. Consider coloring the change in a different color to highlight this difference.

Equation (5): what about physical attacks seems like it should have something in the transitions denoting the changes in states in this case.

The results in figures 1 and 2 are nice. Please clarify in the text and figure captions what the top and bottom rows of figures are

References:
Brown et al. "Bayesian Robust Optimization for Imitation Learning" NeurIPS 2020.
-also considers robust imitation learning but from a different angle.

Zhang et al. "Query-Efficient Imitation Learning for End-to-End Autonomous Driving" and Hoque et al. "LazyDAgger: Reducing Context Switching in Interactive Imitation Learning"
-both consider making DAgger more efficient by not requiring human labels at every state. Could something similar be done with this work?


**Reviewer Expertise:**

Very good: Comprehensive knowledge of the area

**Strengths And Weaknesses:**

strengths:
+novel and interesting problem
+theory justfying method
+nice empirical results

weaknesses:
-no robotics experiments
-no human feedback
-DAgger is not very practical since it requires human labels for every state.
-no empirical comparison with domain randomization as a means to robustify an imitation learning algorithm

**Summary Of Recommendation:**

This paper provides a nice proof of concept for generating and defending against adversarial attacks on an imitation learning agent. I think with improvements to the writing and additional experiments comparing with domain randomization this could be a strong paper and I am willing to improve my recommendation is these things are addressed. Another weakness is the lack of any real robotics experiments or human studies.

---

### Meta-Review · Area_Chair_BhBC · 2021-08-10

**Recommendation:** Accept (Poster)
**Confidence:** 4

**Metareview:**

The authors present a system in which policies are trained to be robust to adversarial changes in the observation / state space. Reviewers consistently praised the novelty of the problem formulation as applied to robotic imitation learning (even if such techniques are common in other ML domains), as well as the presentation of theory justifying the proposed method. The reviewers addressed the bulk of reviewer concerns, adding additional (relevant) baselines and overall increasing the quality and clarity of the submission. Multiple reviewers pointed out lack of real-world experiments as the largest weakness of the paper (even after revisions), however two reviewers pointed out (and I agree) that the paper would still be of interest to the CoRL community despite this.

---

### Decision · Program_Chairs · 2021-09-13

**Decision:**

Accept (Poster)

**Comment:**

The authors present a system in which policies are trained to be robust to adversarial changes in the observation / state space. Reviewers consistently praised the novelty of the problem formulation as applied to robotic imitation learning (even if such techniques are common in other ML domains), as well as the presentation of theory justifying the proposed method. The reviewers addressed the bulk of reviewer concerns, adding additional (relevant) baselines and overall increasing the quality and clarity of the submission. Multiple reviewers pointed out lack of real-world experiments as the largest weakness of the paper (even after revisions), however two reviewers pointed out (and I agree) that the paper would still be of interest to the CoRL community despite this.